# TrkB dependent adult hippocampal progenitor differentiation mediates sustained ketamine antidepressant response

Zhenzhong Ma [1,2], Tong Zang[3], Shari G. Birnbaum[4], Zilai Wang[1,5], Jane E. Johnson[2], Chun-Li Zhang [3] & Luis F. Parada[1,5]

Adult neurogenesis persists in the rodent dentate gyrus and is stimulated by chronic treatment with conventional antidepressants through BDNF/TrkB signaling. Ketamine in low doses produces both rapid and sustained antidepressant effects in patients. Previous studies have shed light on post-transcriptional synaptic NMDAR mediated mechanisms underlying the acute effect, but how ketamine acts at the cellular level to sustain this anti-depressive function for prolonged periods remains unclear. Here we report that ketamine accelerates differentiation of doublecortin-positive adult hippocampal neural progenitors into functionally mature neurons. This process requires TrkB-dependent ERK pathway activation. Genetic ablation of TrkB in neural stem/progenitor cells, or pharmacologic disruption of ERK signaling, or inhibition of adult neurogenesis, each blocks the ketamine-induced behavioral responses. Conversely, enhanced ERK activity via *Nf1* gene deletion extends the response and rescues both neurogenic and behavioral deficits in mice lacking TrkB. Thus, TrkB-dependent neuronal differentiation is involved in the sustained antidepressant effects of ketamine.

[1] Department of Developmental Biology & Kent Waldrep Center for Basic Research on Nerve Growth and Regeneration, University of Texas Southwestern Medical Center, Dallas, TX 75390, USA. [2] Department of Neuroscience, University of Texas Southwestern Medical Center, Dallas, TX 75390, USA. [3] Department of Molecular Biology, University of Texas Southwestern Medical Center, Dallas, TX 75390, USA. [4] Department of Psychiatry, University of Texas Southwestern Medical Center, 5323 Harry Hines Blvd, Dallas, TX 75390, USA. [5] Present address: Brain Tumor Center & Program in Cancer Biology and Genetics, Memorial Sloan Kettering Cancer Center, 1275 York Ave, New York, NY 10065, USA. Correspondence and requests for materials should be addressed to Z.M. (email: zhenzhong.ma@utsouthwestern.edu) or to L.F.P. (email: paradal@mskcc.org)

M ajor depressive disorder (MDD) is a significant cause of disability worldwide, having a devastating impact on both individuals and society[1]. Currently, available antidepressant drugs typically modulate monoaminergic neurotransmission and take several weeks to months to exert their biological effects and achieve clinical efficacy[2]. Moreover, over half of patients with major depression do not have an appreciable response to conventional monoamine re-uptake inhibitor-based

therapy[3,4]. In clinical studies, a single low dose of ketamine, a non-competitive glutamatergic N-methyl-D-aspartate receptor (NMDAR) antagonist, produced both fast-acting and sustained antidepressant effects in patients resistant to conventional antidepressants and at high suicide risk[5–7]. Several mechanisms have been proposed for these acute ketamine antidepressant effects that can be observed within 1 h of administration in rodents. Among the invoked underlying mechanisms are suppression of spontaneous

**Fig. 1** Ketamine accelerates DCX+ progenitor differentiation into NeuN+ newborn neurons in adult DG. **a** The estimated timeline for the development stages of adult hippocampal neurogenesis. Within 2–3 weeks, newborn radial glia-like neural stem/progenitor cells (NPC) could transiently proliferate, survive and differentiate into doublecortin (DCX) positive neuroblasts and immature neurons, which progress into NeuN positive mature neurons at about 28 days after birth. **b** Diagram for the analysis of DCX+ cell differentiation in the wild-type (WT) mice. To allow for BrdU to be incorporated in DCX+ cells, BrdU pulses were administered 18 days before saline (Sal) or ketamine (Ket) treatment, and animals were sacrificed (SAC) either at 24 h or 7 days after the treatment. **c** Representative confocal images of the DG show four types of cells: BrdU+ cells (blue) that co-express strong DCX (green, denoted by solid arrows) or NeuN (red, denoted by arrowheads); cells are labeled with BrdU alone (denoted by empty arrow); BrdU unlabeled DCX+ immature neurons (by asterisk). A magnified image of a typical newborn mature neuron (NeuN+/DCX−/BrdU+) in an orthogonal view is shown on the right. Scale bar, 10 μm for both images. **d** Quantification of NeuN+/DCX− cell percentage in total BrdU+ cells at both 24 h and 1 week time points ($n = 6$ per group for 24 h and $n = 3$ per group for 7 days; BrdU-labeled cell number in the DG is $247.5 \pm 59.55$ vs. $215 \pm 38.59$ for saline vs ketamine at 24 h; $198.7 \pm 12.91$ vs. $184.7 \pm 31.39$ saline vs ketamine at 7 days, respectively; unpaired two-tailed t-test, ***$p < 0.0001$ for saline vs ketamine at 24 h; *$p = 0.0225$, saline vs ketamine at 7 days). **e** Quantification of DCX+ cell percentage of total BrdU+ cells at both 24 h and 1 week time points ($n = 7$ per group at 24 h; $n = 3$ per group at 7 days; unpaired two-tailed t-test, **$p = 0.0046$ for saline vs ketamine at 24 h; **$p = 0.0095$, saline vs ketamine at 7 days). Circles in solid color denote female mouse data points

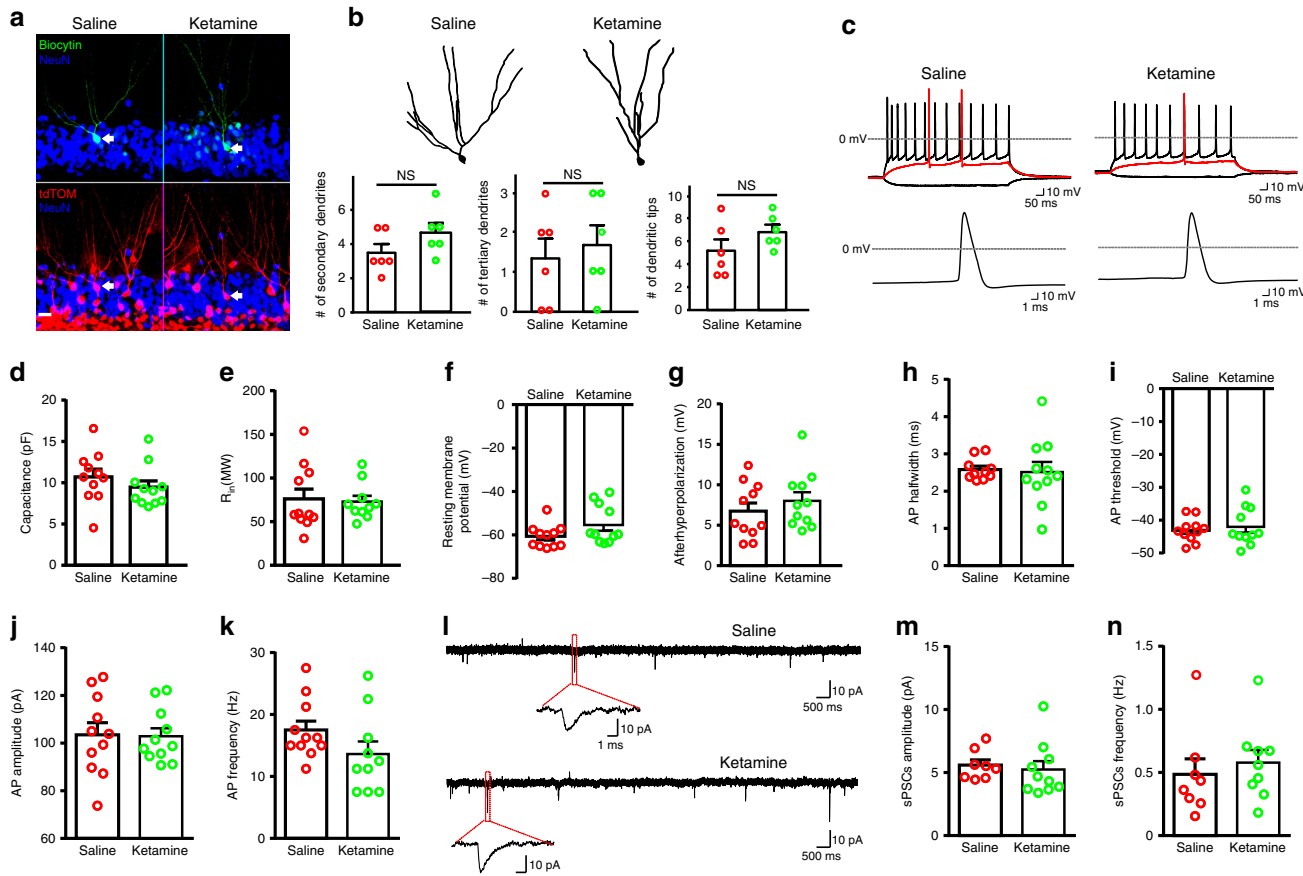

**Fig. 2** Electrophysiology properties of newly differentiated neurons. **a** Representative morphology of patched newborn neurons (NeuN, blue) filled with biocytin (green) and traced by tdTomato (tdTOM, red). Scale bar, 20 μm. **b** Tracing of representative neurons shown in **a** and the quantification of secondary ($p = 0.1504$) and tertiary ($p = 0.6438$) dendrite number as well as the number of dendritic tips ($p = 0.1778$) ($n = 6$ per group; not significant (NS) for all comparisons by unpaired two-tailed $t$-test). **c** Representative action potential (AP) waveforms recorded under current-clamp mode from patched neurons. The precondition sweep and sweeps immediately above threshold (red trace) and at the highest frequency are shown in the upper panel, and a single AP spike is shown in the lower panel. **d** A plot of neuron capacitance under saline or ketamine ($n = 11$ per group; $p = 0.3112$ by unpaired two-tailed $t$-test). **e** A plot of neuron input resistance under saline or ketamine (saline, $n = 11$; ketamine, $n = 10$; $p = 0.5573$ by Mann–Whitney test). **f** A plot of neuron resting membrane potential under saline or ketamine ($n = 11$ per group; $p = 0.1874$ by Mann–Whitney test). **g** A plot of neuron after-hyperpolarization (AHP) under saline or ketamine ($n = 11$ per group; $p = 0.3936$ by unpaired two-tailed $t$-test). **h** A plot of neuron AP half width under saline or ketamine (saline, $n = 10$; ketamine, $n = 11$; $p = 0.8284$ by unpaired two-tailed $t$-test). **i** A plot of neuron AP threshold under saline or ketamine ($n = 11$ per group; $p = 0.8284$ by unpaired two-tailed $t$-test). **j** A plot of neuron AP amplitude under saline or ketamine ($n = 11$ per group; $p = 0.9170$ by unpaired two-tailed $t$-test). **k** A plot of neuron AP frequency under saline or ketamine (saline, $n = 11$; ketamine, $n = 10$; $p = 0.1305$ by unpaired two-tailed $t$-test). **l** Representative traces of spontaneous postsynaptic currents (sPSCs) recorded from differentiated neurons are shown. **m** A plot of neuron sPSCs amplitude under saline or ketamine (saline, $n = 8$; ketamine, $n = 10$; $p = 0.3154$ by Mann–Whitney test). **n** A plot of neuron sPSCs frequency under saline or ketamine (saline, $n = 8$; ketamine, $n = 9$; $p = 0.3213$ by Mann–Whitney test)

neurotransmission mediated NMDAR activation with subsequent enhanced BDNF protein synthesis in the hippocampus[8,9], and rapid synaptic dysfunction recovery through mTOR pathway activation in the prefrontal cortex[10,11]. However, the mechanisms underlying the more extended action of ketamine are not fully understood and remain the subject of intense investigation.

For over a decade, the role of neurotrophic factors in the etiology of depression and the anti-depressive response has been well documented[12]. In rodent models, brain-derived neurotrophic factor (BDNF) and the signaling by its receptor tyrosine kinase, TrkB, have been implicated in stress and mounting the anti-depressive response of selective monoamine reuptake inhibitors. Chronic stress decreases BDNF levels in the hippocampus and causes neuronal atrophy and cell death. In adult nonhuman primates, hippocampal neurogenesis is impaired by stress and can be stimulated by stress coping causing improvement in

antidepressant activity[13,14]. Conversely, chronic antidepressant treatment elevates BDNF expression, supports neuronal survival, and enhances proliferation of neural progenitor cells (NPCs) in the subgranular zone (SGZ) of the dentate gyrus (DG). Enhanced NPC proliferation eventually drives increased neurogenesis and the enhanced behavioral response to antidepressant exposure[12,15,16]. Additional observations implicating hippocampal neurogenesis in mood control include that irradiation-mediated or genetic suppression of adult hippocampal neurogenesis compromises the chronic antidepressant effect and stress responses[17,18]. An increase in new cell generation has also been reported in human depression patients who received antidepressant treatment before death[19]. Our own previous studies have demonstrated that in mouse models, BDNF/TrkB signaling in hippocampal neural progenitor cells (NPCs) regulates behavioral sensitivity to conventional antidepressants[20,21].

Ketamine antidepressant effect does not involve serotonin reuptake inhibition. It is known to stimulate rapid BDNF expression in the hippocampus through a translation mechanism by which phosphorylated eukaryotic elongation factor 2 (eEF2) becomes reduced in both neuronal somata and dendrites[8]. Given that specific subtypes of *Bdnf* mRNA transcripts are locally translated for secretion in an activity-dependent manner in somata[22], ketamine treatment could render BDNF protein accessible to the neurogenic niche of the DG. This raises interesting questions regarding how the rapid biological effects of NMDAR blockade might relate to the more latent actions of monoamine reuptake inhibition, and whether BDNF/TrkB mediated hippocampal neurogenesis may represent a nexus for both mechanisms.

In the present study, we examine the effects of ketamine on hippocampal neurogenesis and find that ketamine accelerates doublecortin (DCX) positive progenitor differentiation into mature neurons within 24 h. Genetic or pharmacologic disruption of DCX+ progenitor differentiation results in a blockade of the sustained behavioral response induced by ketamine. We further identify TrkB signaling and its downstream ERK pathway activation as essential for the generation of newborn neurons and consequent behavioral responses. Moreover, the enhancement of ERK activity by genetically deleting *Nf1* in adult DG progenitors, not only prolongs the ketamine antidepressant response but also rescues both the neurogenic and behavioral deficits in TrkB mutant mice. Our results establish an essential role for BDNF/TrkB-dependent adult hippocampal neurogenesis in maintaining ketamine-mediated antidepressant effects and provide support for the notion that hippocampal neurogenesis is an essential modulator of depression.

## Results

**Ketamine accelerates DG progenitor cell differentiation**. To investigate a potential role for adult hippocampal neurogenesis in mediating ketamine antidepressant effects, we first examined adult hippocampal neurogenesis following administration of the same dose (7 mg/kg) that is sufficient to induce behavioral response[8]. Twenty-four hours after treatment, mice had normal neural progenitor cell (NPC) proliferation and differentiation as measured by Ki67+ and DCX+ cell number respectively (Supplementary Fig. 1a). Analysis at 1 week showed an increased DCX+ progenitor cell number (Supplementary Fig. 1b). In the adult hippocampus, radial glia-like neural stem/progenitor cells (Nestin+) give rise to DCX+ progenitors over a two week period that progress into mature neurons (NeuN+, DCX−) in the third week (Fig. 1a). A BrdU pulse-chase experiment to label proliferating NPCs (Supplementary Fig. 1c) revealed that the ketamine increased BrdU incorporation was not present in the DCX positive newborn cells (DCX+/BrdU+) 1 week after treatment (Supplementary Fig. 1d, e). Instead as shown below, the BrdU was prematurely chased into the newborn neuron (NeuN) population. We saw no evidence of enhanced NPC proliferation or DCX+ progenitor cell number indicating that early phases of neurogenesis were unaffected. To better examine the late phase of neurogenesis, we next performed a BrdU pulse to label the proliferating neural progenitors and chased them for either 19 or 25 days with ketamine exposure at day 18 (Fig. 1b). The data show that one dose of ketamine accelerated newborn neuron (NeuN+, DCX−) generation among the BrdU+ cells at both 24 h (day 19) and 1-week (day 25) post-treatment (Figs 1c, d). Correspondingly, the increase in newborn neurons was accompanied by a decrease in DCX+ cells among BrdU+ cells at both time points (Fig. 1e). Accelerated differentiation of DCX+ progenitors to NeuN+ mature neurons was also observed when we traced the

nestin+ neural stem/progenitor cell lineage in the DG with a Rosa26-tdTOM reporter (Supplementary Fig. 1f–i), driven by a tamoxifen-inducible Nestin-creER$^{T2}$ transgene that is highly efficient and specific in targeting the vast majority of neural stem/progenitor cells in the DG[21,23] (Supplementary Fig. 2a). Taken together, these data indicate that ketamine exerts no effects on quiescent stem cell activation, or transition to DCX+ cells, but instead promotes accelerated differentiation of existing DCX+ progenitors into newborn neurons.

**Newborn neurons are functionally mature**. To examine whether newborn neurons resulting from ketamine treatment are functional and integrate into the existing DG neural circuitry, we studied their electrophysiological properties by taking whole-cell patch clamp recordings. Recorded neurons were traced by using Nestin-CreER$^{T2}$ –tdTomato to ensure they originated from NPCs in the DG (Fig. 2a). By morphology, newborn neurons following ketamine treatment display normal dendritic branching, including secondary/tertiary dendrites and dendritic tips, compared with controls (Fig. 2b). All recorded newborn neurons fired repetitive action potentials (APs) upon current injection when examined at 24 h after saline or ketamine treatment (Fig. 2c), and had inward and outward currents in voltage-clamp mode, through voltage-gated sodium and potassium channels, respectively (Supplemental Fig. 2e). The neurons from saline and ketamine treatment showed similar intrinsic cell properties, including capacitance, input resistance, and resting membrane potential (Figs 2d–f). They also showed similar excitability, including AP after hyperpolarization, half width, threshold, amplitude, frequency (Figs. 2g–k) and delay of the first spike, the maximum velocity of rising and decay (Supplemental Fig. 2b–d). There were no significant differences in channel properties among these differentiated neurons, as measured by sodium and potassium current amplitudes (Supplemental Fig. 2f–h). Importantly, typical spontaneous postsynaptic currents (sPSCs) were detected, indicating functional synapse formation between these neuron populations and local neural circuitry (Fig. 2l). Saline or ketamine treatment had similar frequency and amplitude of sPSCs, suggesting the comparable synapse connectivity when ketamine-induced newborn neurons integrated into the local network (Figs. 2m, n). Thus, morphological and electrophysiological analysis of ketamine exposed newborn hippocampal neurons indicated normal differentiation and activity.

We further assessed the response of ketamine exposed newborn neurons to pathological states by the induction of seizures. The chemoconvulsant, kainic acid (KA), is a potent agonist of glutamate kainate receptors and systemic administration induces seizures with classic immediate early gene transcription response in the hippocampus[24,25]. Thus, newborn neurons that incorporate into the hippocampal circuitry will respond to the KA stimulated neuro-excitation wave and exhibit an immediate early gene response including the well-characterized immediate early response gene, EGR1 that is a marker of the maturity of adult-born hippocampal granule neurons[26,27]. Mice were subjected to BrdU pulses and chased for 18 days to allow time for BrdU presence in late stage progenitors and newborn neurons. Two hours after KA was administered, mice were killed and assayed for co-expression of BrdU and EGR1 (Supplementary Fig. 3a). KA induction leads to EGR1 activation in hippocampal neurons in all animals, exhibiting a similar percentage of triple positive cells (>80%; BrdU+/NeuN+/EGR1+) in newborn neurons (BrdU+/NeuN+) (Supplementary Fig. 3d, e, f). However, consistent with the finding that ketamine accelerates newborn neuron generation (Fig. 1), ketamine-treated animals exhibited an increased number of EGR1

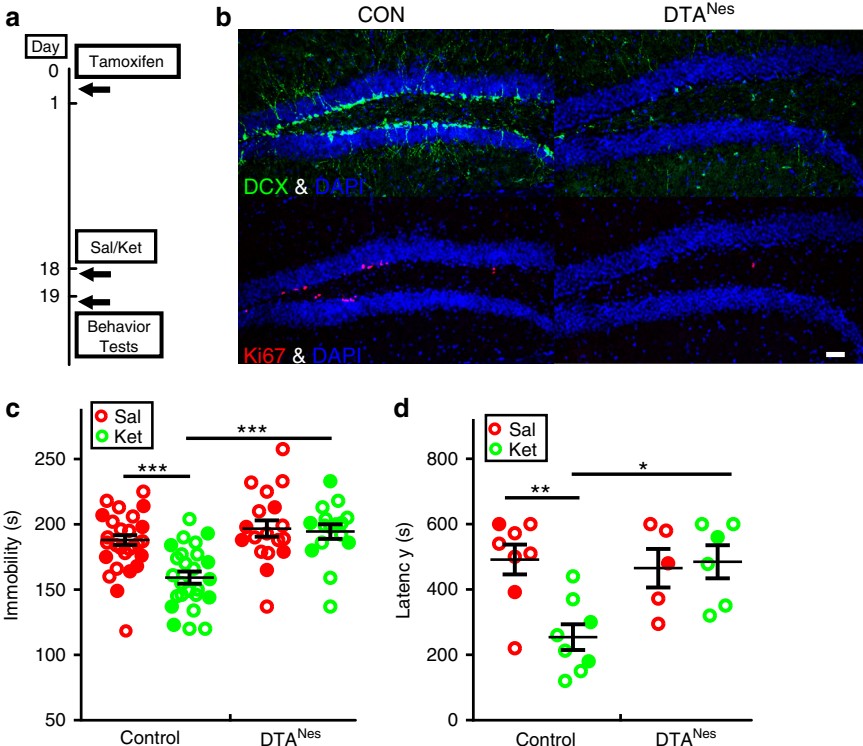

**Fig. 3** DTA mediated ablation of DCX + cell population abolishes the ketamine antidepressant effect. **a** Nestin-creER$^{T2}$; Rosa26-DTA (DTA$^{Nes}$) heterozygote mice were given tamoxifen 18 days before Saline (Sal) or Ketamine (Ket) treatment and were exposed to behavior tests 24 h afterward. **b** Representative confocal images show the efficient elimination of adult neural stem/progenitor cells, as measured by DCX (green) and Ki67 (red). Scale bar, 50 μm. **c** In the forced swim test (FST), the significant decrease of immobility time induced by ketamine treatment is fully blocked in DTA$^{Nes}$ mice (2-way ANOVA with unweighted mean analysis, Tukey post hoc,***$p = 2.063E-5$ for ketamine vs. saline-treated control mice and ***$p = 5.471E-6$ ketamine-treated control mice vs. DTA$^{Nes}$ mice). **d** DTA$^{Nes}$ mice treated with ketamine do not display a reduction in latency to feed in the novelty-suppressed feeding test (NSFT; two-way ANOVA, Tukey post hoc **$p = 0.0041$ for saline vs. ketamine-treated control animal; *$p = 0.0105$ for ketamine-treated control mice vs. DTA$^{Nes}$ mice). Circles in solid color denote female mouse data points

expressing newborn neurons (BrdU+/NeuN+/EGR1+) (Supplementary Fig. 3b, c). Thus, by the above criteria, newborn neurons arising from ketamine accelerated progenitor differentiation are functionally mature and exhibit similar synaptic activation as normal newborn neurons in both physiologic and pathologic states.

**DCX+ progenitor differentiation sustains ketamine effect.** To further explore the physiologic relevance of accelerated DCX+ to NeuN+ cell differentiation in the behavioral response to ketamine, we selectively eliminated Nestin+ neural stem cells and their immediate progeny prior to ketamine exposure using a floxed-STOP-diphtheria toxin subunit A (DTA) transgene embedded within the Rosa26 locus that was activated by Nestin-creER$^{T2}$ transgene (DTA$^{Nes}$)[28]. Adult DTA$^{Nes}$ mice were treated with tamoxifen to activate DTA expression followed by saline or ketamine treatment eighteen days later (Fig. 3a). This eighteen-day induction paradigm restricts DTA elimination to stem cells and progenitors that would have normally been produced within the eighteen-day period but minimally affects newborn or mature neurons that were formed prior to the induction. Nestin-creER$^{T2}$ induced recombination is restricted to the neurogenic niches (SGZ and SVZ) in adult brain (Supplementary Fig. 4), and thus DTA$^{Nes}$ tamoxifen-treated mice did not display cell loss in other brain regions and were indistinguishable from their littermate controls in brain morphology, brain weight and structures (Supplementary Fig. 5). However, adult neurogenesis was efficiently eliminated, with complete loss of Ki67 positive proliferating NPCs and dramatic reduction of DCX+ cell number

(5% remaining compared to control) in the DG (Fig. 3b). Pre-existing mature neurons and astrocytes appeared to be unaffected in DTA$^{Nes}$ mice (Supplementary Fig. 5d–g). Moreover, DTA$^{Nes}$ neurogenesis-deficient mice were rendered insensitive to the antidepressive effects of ketamine 24 h after treatment as observed in both the forced swim test (FST) and novelty-suppressed feeding test (NSFT) (Figs. 3c, d). These data indicate that the late-stage progenitor (DCX+) pool normally available for conversion into mature neurons, but eliminated in this paradigm by DTA, is essential for the sustained behavioral response to ketamine.

We also used a pharmacologic approach to target proliferating neural progenitor cells in the hippocampus. Temozolomide (TMZ) is a DNA alkylating agent in use to treat patients suffering from glioblastoma multiforme where it targets the BrdU-incorporating (dividing) tumor cell population and transiently arrests tumor growth[29]. This potent chemotherapeutic agent has also been shown to effectively impair adult hippocampal neurogenesis in mouse models of learning and memory[30]. As shown in Supplementary Fig. 6, TMZ impairs adult hippocampal neurogenesis, as measured by both Ki67 and DCX. To avoid any potential confounding side effects of acute TMZ treatment on behavior assays, mice were subjected to four TMZ cycles followed by a 1-month gap prior to behavioral testing (Fig. 4a). This paradigm results in the selective reduction of late-phase DCX+ progenitors. After termination of TMZ treatment, recovery of the progenitor population is mediated by quiescent cell re-entry into the cell cycle. Indeed, the TMZ-treated mice have the same number of total DCX+ cells as controls, but are overrepresented by early progenitors (DCX+/Ki67+) compared to controls

(Figs. 4b, d, e). Consistent with the experimental design, the DCX + progenitor population following TMZ treatment exhibits properties of newly formed immature progenitors including reduced dendritic processing and proximity to their source in the subgranular zone (SGZ). In contrast, vehicle treated mice have abundant numbers of late phase progenitors (56.0% ± 2.41 vs 23.2% ± 3.05) with established dendritic morphology extending through the granular layer zone (GLZ) into the molecular layer (Figs. 4b, c and f). These data are consistent with the model that TMZ treatment results in a greatly reduced source of the late-phase progenitors for the production of newborn neurons. Upon treatment arrest, quiescent stem cells re-enter the cell cycle aggressively to compensate for the early progenitor loss caused by the drug exposure. As shown in Fig. 4g, although both vehicle and

TMZ-treated mice respond to ketamine by accelerating newborn neuron production, TMZ mice have a markedly impaired capacity and only exhibit about 20% of new neurons generated in ketamine-treated vehicle mice. Consistent with the previous DTA experiment, the consequence of TMZ treatment and progenitor cell destruction is a blockade of the ketamine anti-depressive behavior effect in FST (Fig. 4h). Taken together, the genetic lineage ablation and pharmacological studies indicate that ketamine exerts its sustained anti-depressive behavior by accelerating the formation of newborn neurons from the pre-existing pool of mature DCX+ progenitors.

**TrkB mediates ketamine antidepressant effect**. In the hippo-campus, ketamine accelerates DCX+ progenitor differentiation

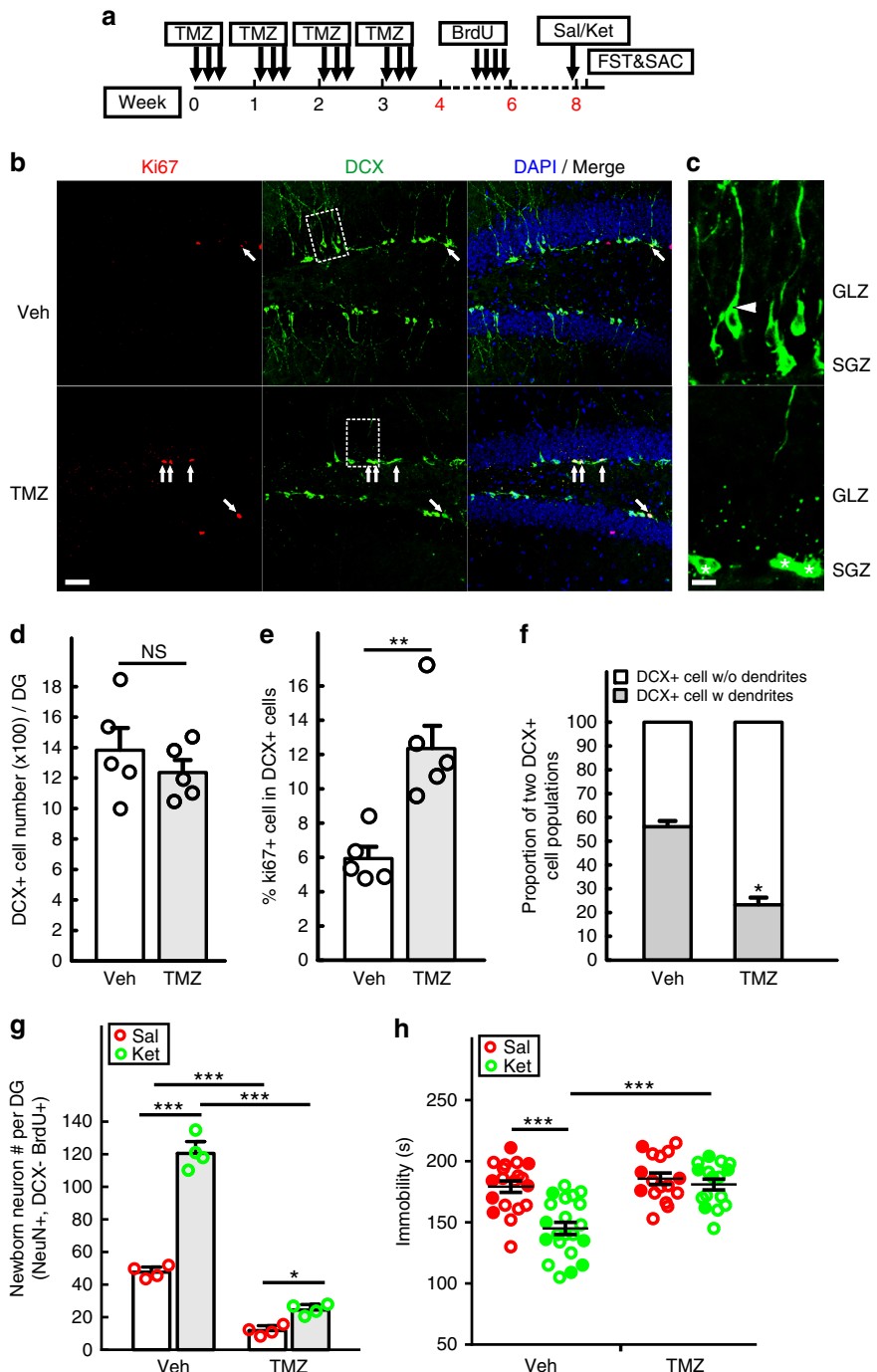

within 24 h (Fig. 1), which correlates with the rapid and significant increase in BDNF protein levels observed in the DG (Fig. 5a). Previous studies have shown that BDNF plays an important role in NPC survival, proliferation, differentiation and the behavioral response to chronic treatment with conventional antidepressants[20,21,31,32]. TrkB, the primary receptor for BDNF, is expressed in the entire neural progenitor lineage in adult mouse DG[20,33]. Therefore we tested a possible role for TrkB in the ketamine-induced acceleration of DCX+ progenitor differentiation and the anti-depressive behavioral response. Using Nestin-creER[T2] mice, we specifically ablated *TrkB* (TrkB flox/flox) in the adult neural progenitor lineage (TrkB[Nes]) and exposed the TrkB deficient mice to different anti-depressive behavioral tests six weeks after tamoxifen induction (Supplementary Fig. 7a). Although TrkB ablation in the adult DG did not alter the basal level of depression-like behavior in mice[20], the ketamine anti-depressant behavioral effect was fully blocked as measured by immobility in the FST and feeding latency in the NSFT (Supplementary Fig. 7b, c). By the end of the NSFT (10 min), 81.8% of ketamine-treated control mice had fed compared with 66.7% of TrkB[Nes] f/f mice (Supplementary Fig. 7d). All mice consumed similar food quantity following the latency test, indicating that the behavioral change was not caused by altered appetite or motivation to feed (Supplementary Fig. 7e). In the learned helplessness test, the ketamine-treated TrkB[Nes] mice escaped in fewer trials compared with ketamine-treated control animals ($7.85 \pm 1.22$ vs $11.46 \pm 0.89$) (Supplementary Fig. 7f). Also, unlike control animals that responded to ketamine and displayed significantly shorter escape latencies, TrkB[Nes] mice had an attenuated ketamine response at the beginning of the test (trials 4–9 of total 15 trials) but did eventually learned to escape from the chamber (Supplementary Fig. 7g). The data indicate that TrkB is critical in adult newborn neurons for mounting ketamine anti-depressive behavioral response.

To further investigate the role of TrkB in mediating DCX+ progenitor differentiation and ketamine action, we employed a 3-week post tamoxifen induction paradigm to target TrkB ablation to NPCs while minimizing TrkB loss in newborn mature granule neurons (Fig. 5b). As anticipated, control mice exhibit accelerated neuronal differentiation 24 h after ketamine treatment. In contrast, mice with TrkB deleted NPCs had impaired generation of newborn neurons and instead an increased percentage of DCX + progenitors compared with ketamine-treated control mice (Figs 5c, d). Therefore, ketamine mediates increased total number of newborn neurons in the DG that can be blocked by TrkB ablation (Supplementary Fig. 7h). Accordingly, TrkB mutant mice displayed significantly increased immobility time in the FST compared with control mice at both 24 h and 1 week after

ketamine treatment (Figs. 5f, g). Similar results were seen in the NSFT where feeding latency was not affected by ketamine treatment in TrkB[Nes] mice (Fig. 5h). Moreover, cumulative feeding latency analysis revealed that when 100% of ketamine-treated control mice had fed, only 31% of ketamine-treated TrkB[Nes] mice had fed (Fig. 5I). This difference cannot be attributed to altered appetite because all groups of mice consumed food equally with equivalent feeding behavior (Fig. 5J). Importantly, by virtue of the 3-week tamoxifen paradigm, these TrkB mutant mice retained the acute (1 h) anti-depressive response to ketamine (Fig. 5e), which is mediated by fast-acting synaptic mechanisms in pre-existing mature neurons that are not affected by this paradigm[8,10]. Thus sustained ketamine behavioral response as well as the accelerated progenitor differentiation requires functional TrkB.

**TrkB-dependent signal transduction regulates ketamine effect**. To elucidate the mechanism by which hippocampal progenitor differentiation is accelerated by ketamine-induced BDNF/TrkB signaling, we examined the major effector pathways, that have been implicated in BDNF/TrkB signaling[21,22,34,35]. Given that ketamine has a short half-life (~3 h)[8] and can increase BDNF protein levels within 1 h (Fig. 5a), we examined early responses of signaling pathways by measuring phosphorylation levels of TrkB, ERK (RAS/MAPK pathway), AKT (PI3K pathway) and S6 (mTOR pathway) in the DG. We observed enhanced ERK phosphorylation, but not AKT or S6 after ketamine treatment (Fig. 6a). Importantly, in TrkB[Nes] DCX+ progenitors, p-ERK was equivalent to controls (Supplementary Fig. 9a), consistent with a requirement for BDNF/TrkB signaling to induce ERK activation. We further observed that immature DCX+ progenitors had relatively higher ERK phosphorylation than more mature progenitors (Fig. 6b), indicating that DCX+ progenitors have differential ERK activation response to ketamine. Ketamine also induced ERK mild activation in the CA1, but not in the CA3 and cerebellum (Supplementary Fig. 9b). Western blot analysis from isolated DG tissue confirmed that ketamine-induced a TrkB-dependent activation of the ERK pathway (Figs 6c, d). Interestingly, ERK activation seems to be transient in mature hippocampal neurons. ERK phosphorylation disappeared in the granular layer but was maintained in neural progenitors 24 h after ketamine treatment (Supplementary Fig. 9c).

To directly test the role of ERK pathway activation in mediating hippocampal progenitor differentiation and the behavioral response to ketamine, we pretreated mice with a selective MEK inhibitor, SL327[36], 1 h before ketamine treatment (Fig. 7a). This pre-treatment suppressed ketamine-mediated ERK activation by 25% as measured by normalized ERK

**Fig. 4** Temozolomide mediated suppression of adult neurogenesis impairs ketamine antidepressant effect. **a** A timeline for the suppression of adult hippocampal neurogenesis by four cycles of temozolomide (TMZ). Each cycle includes single daily injections (25 mg/kg) on three consecutive days followed by a recovery period of four days. Ten days after the last cycle of TMZ treatment, mice were given four pulses of BrdU to label proliferating NPCs that were born after TMZ exposure. Eighteen days after BrdU pulses, mice were given Sal or Ket and then examined for anti-depressive like behavior in the FST 24 h later. **b, c** Sections from TMZ and vehicle (Veh) treated mice were immunostained with DCX (green), Ki67 (red) and DAPI (blue) for nuclei. DCX + cells that co-express Ki67 are denoted by arrows in the confocal images. Scale bar, 50 μm. High power images of the area from the dash-line rectangle are shown in **c**, to highlight DCX+ cell morphology changes after TMZ treatment. Cells with dendrites (arrowheads) have migrated into the inner granular layer zone (GLZ), while cells without processes (asterisks) are still localized in the subgranular zone (SGZ). Scale bar, 10 μm. **d** Quantification of total DCX + cell number in the DG ($n = 5$ per group, $p = 0.4073$ by unpaired two-tailed *t*-test). **e** Percentage of DCX+ cells that express Ki67 in the DG ($n = 5$ per group, \*\*$p = 0.0027$ by unpaired two-tailed *t*-test). **f** The proportion of DCX+ cells with or without dendrites was quantified in the DG ($n = 5$ per group, \*\*\*$p < 0.0001$ by unpaired two-tailed *t*-test). **g** Quantitative analysis of BrdU pulse-chase for 18 days shows the number of newborn neurons (BrdU+/NeuN +/DCX−) in the DG was significantly decreased (two-way ANOVA, Tukey post hoc, \*\*\*$p < 0.0001$ for saline vs ketamine treatment in vehicle mice, and Veh vs TMZ group with saline or ketamine treatment; \*$p = 0.0444$ for saline vs ketamine treatment in TMZ mice). **h** FST shows ketamine's antidepressant effect is fully abolished in TMZ-treated mice (2-way ANOVA with unweighted mean analysis, Tukey post hoc, \*\*\*$p = 1.299E-6$ for saline vs ketamine in vehicle treated mice and \*\*\*$p = 1.192E-6$ for ketamine-treated mice with vehicle vs TMZ). Circles in solid color denote the data points from female mice

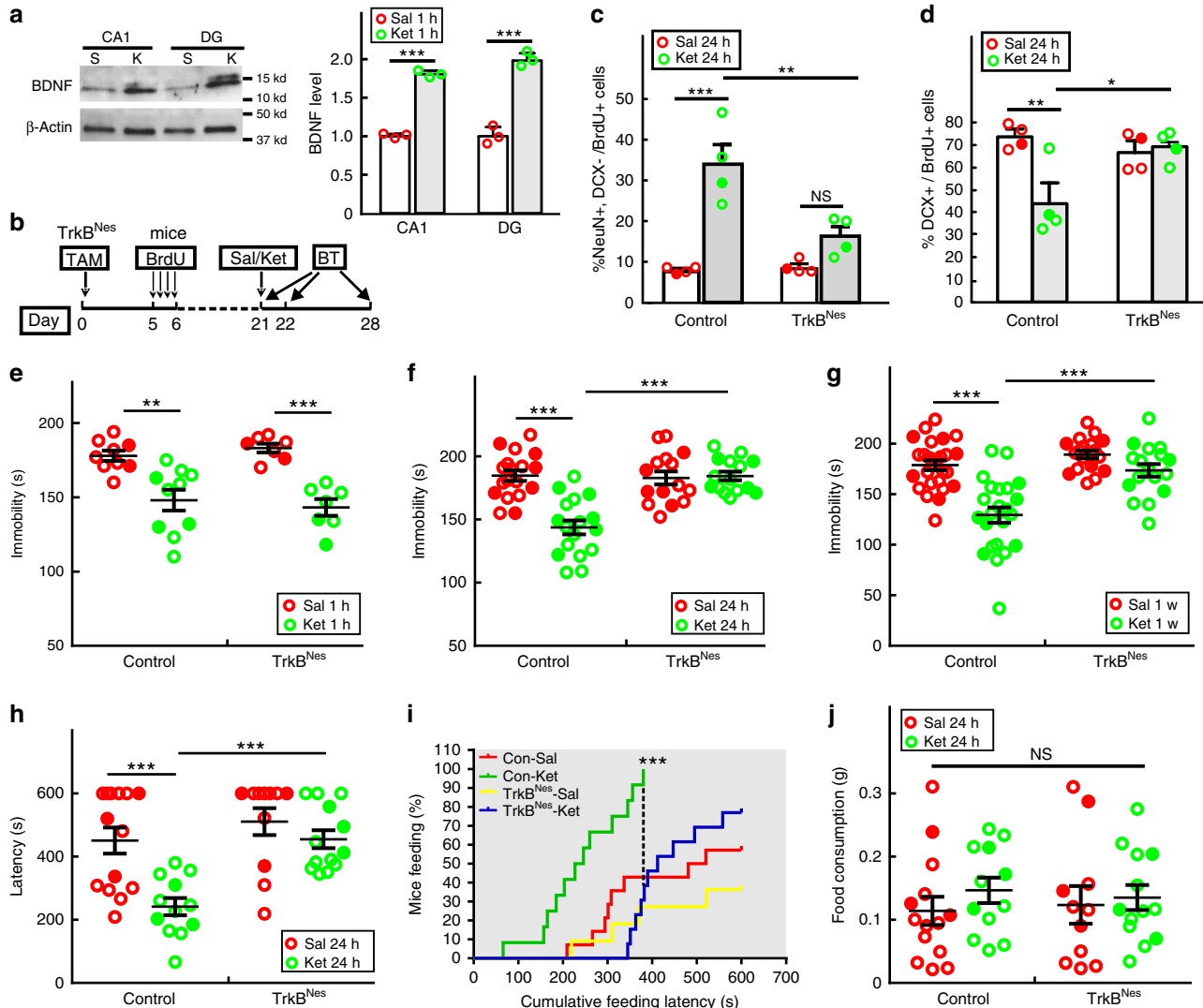

**Fig. 5** TrkB is required in adult NPCs to mediate the antidepressant effects of ketamine. **a** The quantification on BDNF immunoblots of dissected adult CA1 and DG regions shows the BDNF protein level significantly increases 1 h after saline (S) or ketamine (K) treatment. ($n = 3$; ***$p < 0.0001$ for CA1, ***$p < 0.0001$ for DG). **b** Nestin-creER$^{T2}$; TrkB flox/flox (TrkB$^{Nes}$) mice were induced with tamoxifen (TAM) three weeks before saline (Sal) or ketamine (Ket) treatment. BrdU pulses were administrated sixteen days before treatment, and mice were given behavior tests (BT) 1 h, 24 h or 1 week after treatment. **c** The percentage of newborn NeuN+/DCX− neurons in BrdU+ cells (two-way ANOVA, Tukey post hoc, ***$p = 0.0001$ for saline vs ketamine in control mice; **$p = 0.0031$ for ketamine-treated control vs TrkB$^{Nes}$; $p = 0.2510$ for saline vs ketamine in TrkB$^{Nes}$). **d** Percentage of newborn DCX+ cells (two-way ANOVA, Tukey post hoc, **$p = 0.0048$ for saline vs ketamine in control; *$p = 0.0219$ for ketamine treatment in control vs TrkB$^{Nes}$). **e** In the FST, TrkB$^{Nes}$ mice respond to ketamine 1 h after treatment (two-way ANOVA, Tukey post hoc, **$p = 0.0013$ for saline vs ketamine in control mice and ***$p < 0.001$ for saline vs ketamine in TrkB$^{Nes}$). However, they do not retain the behavioral response at both (**f**) 24 h (two-way ANOVA, Tukey post hoc, ***$p < 0.001$ for saline vs ketamine in control mice and for ketamine treatment in control vs TrkB$^{Nes}$) and (**g**) 1 week (two-way ANOVA with unweighted mean analysis, Tukey post hoc, ***$p = 1.773E-8$ for saline vs ketamine in control mice and ***$p = 2.633E−6$ for ketamine treatment in control vs TrkB$^{Nes}$). (**h**) In the NSFT, the TrkB$^{Nes}$ mice have a longer latency to feed compared with ketamine-treated control mice (two-way ANOVA, Tukey post hoc, ***$p < 0.001$ for saline vs ketamine in control mice ($p = 0.0007$) and for ketamine treatment in control vs TrkB$^{Nes}$ ($p = 0.0006$)). **i** Cumulative feeding latency is measured (Mantel–Cox log-rank test ***$p < 0.0001$ for ketamine treatment in control mice compared with other groups). Dash-line highlights the time-point when all ketamine-treated control mice had eaten, compared with a significant lower percentage of mice in the other groups. **j** We observed no confounding effects on appetite (two-way ANOVA). Circles in solid color denote female mouse data points

phosphorylation from dissected DG (Supplementary Fig. 8a, b). One day after ketamine treatment, we subjected mice to FST. One dose of SL327, 24 h prior to behavioral testing did not alter the depression-like behavior in saline-treated mice[8]; However, SL327 pretreatment blocked the ketamine antidepressant effect (Fig. 7c), similar to that observed in TrkB mutant mice. Moreover, the generation of newborn DG neurons was also impaired as demonstrated by the reduction of BrdU+ cells expressing NeuN in the SL327 treated cohort (Fig. 7b). These data are consistent with our model that TrkB-dependent ERK activation is required for ketamine-induced progenitor differentiation and antidepressant response.

A prediction from the preceding studies would be that enhanced or prolonged ERK activation in DG NPCs should promote both the neurogenic and behavioral responses to ketamine. To enhance ERK activation specifically in adult NPCs, we turned to a genetic approach. The protein encoded by the NF1 tumor suppressor, neurofibromin, is a demonstrated negative

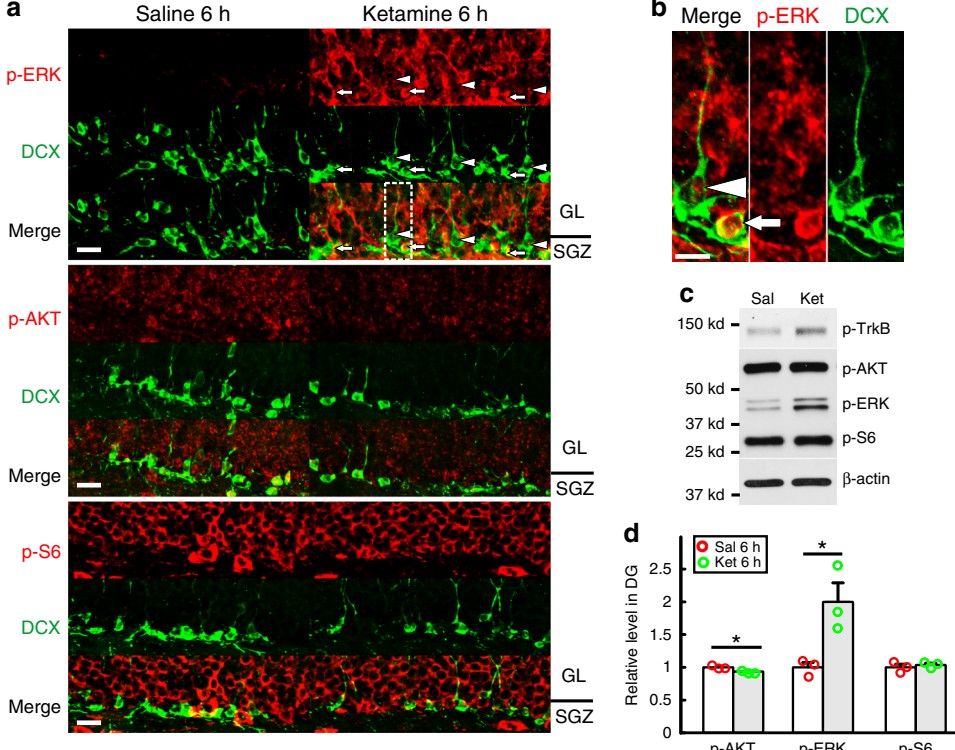

**Fig. 6** Ketamine enhances ERK signaling pathway. **a** Confocal images show immuno-signal of p-ERK (red, upper panel), but not p-Akt (red, middle panel) or p-S6 (red, bottom panel), increases in both granular layer (GL) and subgranular zone (SGZ) of the DG six hours after one dose of ketamine treatment. Arrows denote the DCX+ cells (green) that are localized in SGZ and have a high level of p-ERK. Arrowheads highlight the DCX+ cells with long dendrites that have a modest increase of p-ERK. Scale bar, 20 μm. **b** The image in the dash-line rectangle in (a) is magnified showing a cell without dendrites (arrow) and one with long dendrites (arrowhead). They have varying ERK activation in response to ketamine. Scale bar, 20 μm. **c**, **d** Immunoblots on p-TrkB, p-Akt, p-ERK and p-S6 in dissected DGs from mice after 6 h of ketamine treatment. The quantification of expression level for each protein, normalized by β-actin expression level, is shown in **d** ($n = 3$ per group; $*p = 0.0327$ for p-AKT, $*p = 0.0286$ for p-ERK, $p = 0.5336$ for p-S6 by unpaired two-tailed $t$-test)

regulator (RAS-GAP) downstream of the Trk family of receptor tyrosine kinases[21,34,37]. We used Nes-creER[T2] to genetically delete *Nf1* in adult hippocampal stem cells and their progeny (NF1[Nes]). Adult NF1[Nes] mice were treated with tamoxifen to delete *Nf1* three weeks prior to saline or ketamine administration (Fig. 7d). Twenty-four hours later, the analysis of neurogenesis revealed that NF1[Nes] mice exhibited enhanced generation of hippocampal newborn neurons comparable to the ketamine effect on control mice (Fig. 7e). We also observed that in the absence of NF1, ketamine had no additional effect on newborn neuron production indicating a saturation of neuronal differentiation. Similar to what we have previously reported, DCX+ progenitors with NF1 loss mediated enhanced p-ERK level (Supplementary Fig. 8c), continuously produce adult-born neurons in mutant mice[21]. Behavioral tests four weeks after ketamine treatment indicated that both NF1[Nes] heterozygous and homozygous mice, but not control mice, maintained the antidepressant effect of ketamine (Fig. 7f). Thus pharmacologic or genetic manipulation to activate the ERK signal transduction pathway in NPCs improves the behavioral anti-depressive response to ketamine.

**Enhanced ERK pathway rescues phenotype in TrkB mutants.** To close the circle, given the inhibition of cellular and physiological response to ketamine after genetic TrkB loss and the acceleration following ERK activation, we directly examined epistasis between NF1 and TrkB. Nestin-creER[T2] was combined with both the TrkB and NF1 floxed alleles (TrkB[Nes]; NF1[Nes]). In the same paradigm used for TrkB mutants (Fig. 5b), three weeks

after dual TrkB and NF1 ablation in hippocampal progenitors, mice were subjected to the FST behavioral assay. Consistent with previous results, ketamine induced a decreased immobility time in control, double heterozygote mutant, and NF1 mutant mice, but not in TrkB mutant mice. In contrast, the inclusion of one floxed allele of *Nf1* (TrkB[Nes]; NF1[Nes]) was sufficient to rescue the impaired behavioral response to ketamine (Fig. 8a). Moreover, at the cellular level, *Nf1* ablation successfully reversed the low p-ERK expression in DCX+ cells observed in TrkB[Nes] mice following ketamine treatment (Supplementary Fig. 9a). Commensurately, the deficiency of accelerated neuronal differentiation present in TrkB[Nes] mice was also rescued by co-introduction of an NF1 knock-out allele. A significantly higher percentage of newborn neurons was present in TrkB[Nes]; NF1[Nes] double-mutant mice compared with ketamine-treated TrkB[Nes] mice (Fig. 8b). Consequently, the abolished enhancement of newborn neuron generation by TrkB ablation could be maintained in the ketamine-treated TrkB[Nes]; NF1[Nes] double-mutant mice (Fig. 8c). These results demonstrate a linear signaling relationship between TrkB and ERK pathway function in mediating both hippocampal progenitor differentiation and mounting a behavioral effect in response to acute ketamine induction.

**Discussion**
The continuous generation of new neurons by NPCs in the dentate gyrus provides structural plasticity in the adult dentate gyrus allowing for behavioral response to environmental stimuli, including antidepressants. It has been demonstrated that the

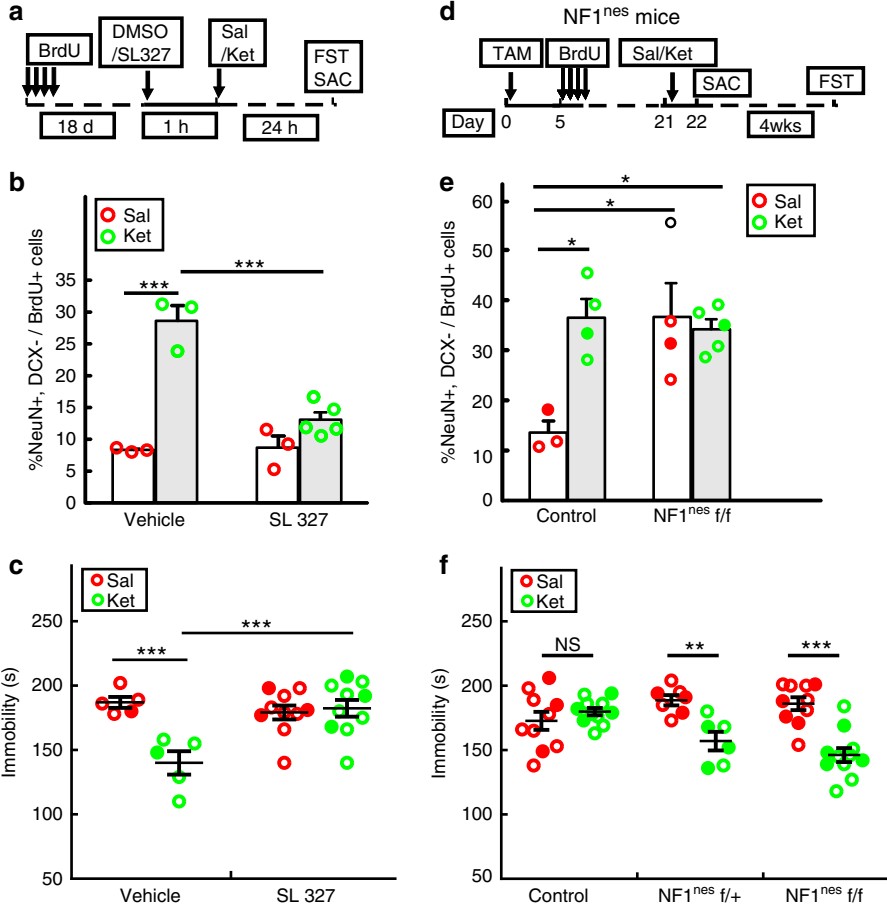

**Fig. 7** ERK pathway regulates the anti-depressive effect of ketamine. **a** The MEK inhibitor, SL327, was administrated to mice 1 h before saline (Sal) or ketamine (Ket) treatment and FST was tested 24 h later. Newborn cells were labeled with BrdU 18 days prior to the drug treatments. **b** Ketamine significantly increases the percentage of newborn neurons only in vehicle treated mice, but not in mice pre-treated with the MEK inhibitor (BrdU+ cell number quantified in the DG is $180 \pm 4.3$, $135.3 \pm 13.3$, $215.6 \pm 7.9$, $236.1 \pm 32.3$, respectively; two-way ANOVA, Tukey post hoc, ***$p < 0.001$ for saline vs ketamine in vehicle mice and for ketamine-treated vehicle vs SL327). **c** Mice pretreated with SL327 do not exhibit reduced immobility time in the FST following ketamine treatment (two-way ANOVA with unweighted mean analysis, Tukey post hoc, ***$p = 3.46E{-}4$ for saline vs ketamine in vehicle mice and ***$p = 2.21E{-}4$ for ketamine-treated vehicle vs SL327). **d** Diagram of treatments for Nestin-creER[T2]; NF1 flox/flox (NF1[Nes]) mice. One group of mice was sacrificed (SAC) 24 h after saline or ketamine treatment for analysis of adult hippocampal neurogenesis. Another group of mice received the FST four weeks after treatment. (**e**) NF1[Nestin] homozygous (f/f) mutant mice have a significant increase in the percentage of newborn neurons. This enhancement is comparable to the level of newborn neurons in ketamine-treated control mice (BrdU+ cell number quantified in the DG is $124.8 \pm 16.9$, $160.2 \pm 10.9$, $167.4 \pm 7.3$ and $173.9 \pm 22.3$, respectively; two-way ANOVA, Tukey post hoc, *$p < 0.05$ for saline-treated control mice vs ketamine-treated control ($p = 0.0163$), saline-treated NF1[Nestin] ($p = 0.0156$) and ketamine-treated NF1[Nestin] ($p = 0.0236$)). (**f**) Immobility time in the FST remains decreased in NF1[Nestin] heterozygous (f/+) and NF1[Nestin] homozygous (f/f) mutant mice but not in control mice 4 weeks after ketamine treatment (two-way ANOVA with unweighted mean analysis, Tukey post hoc, **$p = 0.0011$ for saline vs ketamine in NF1[Nestin] f/+ mice; ***$p = 1.022E{-}6$ for saline vs ketamine in NF1[Nestin] f/f; not significant (NS) for saline vs ketamine in control mice ($p = 0.3183$)). Circles in solid color denote the data points from female mice

almost immediate acute, behavioral response to ketamine can be explained by rapid synaptic plasticity changes and circuit rewiring[9]. Our study provides a functional link between adult hippocampal neurogenesis and the more sustained anti-depressive response to ketamine in mice. By extension and analogy, it provides support for the notion that adult hippocampal neurogenesis is a critical biological determinant of the anti-depressive response[12]. Consistent with previous studies showing that in the hippocampus, BDNF plays a central role in neuronal differentiation and maturation[20,22,24], this work demonstrates that ketamine-mediated production of BDNF accelerates DCX+ progenitor differentiation into mature granule neurons through TrkB signaling and downstream ERK pathway activation. This cellular effect elicits a behavioral anti-depressive response within 24 h and the production of newborn progenitors and neurons seems to last for about 1 week. Specific loss of neural progenitor cells in the

DTA[Nes] and TMZ paradigms significantly impaired the sustained ketamine behavioral response, but not the depression-like behaviors at the baseline that requires newborn mature neurons[18]. We also demonstrate that ketamine acceleration of DCX+ progenitors results in functional newborn neurons that likely serves to modulate the hippocampal neural network to sustain the anti-depressive effect analogous to that observed for chronic monoamine treatment[12].

A recent related study in rats reported that ketamine can promote rapid maturation of hippocampal newborn neurons as defined by the immediate early *EGR1* (zif268) gene response, leading to the generation of more neurons within hours in both normal conditions and in a depression model[25]. This study also reported that the acute anti-depressive effect (1 h) is independent of newborn neurons, which is consistent with our finding that enhanced neurogenesis is required for the sustained antidepressant

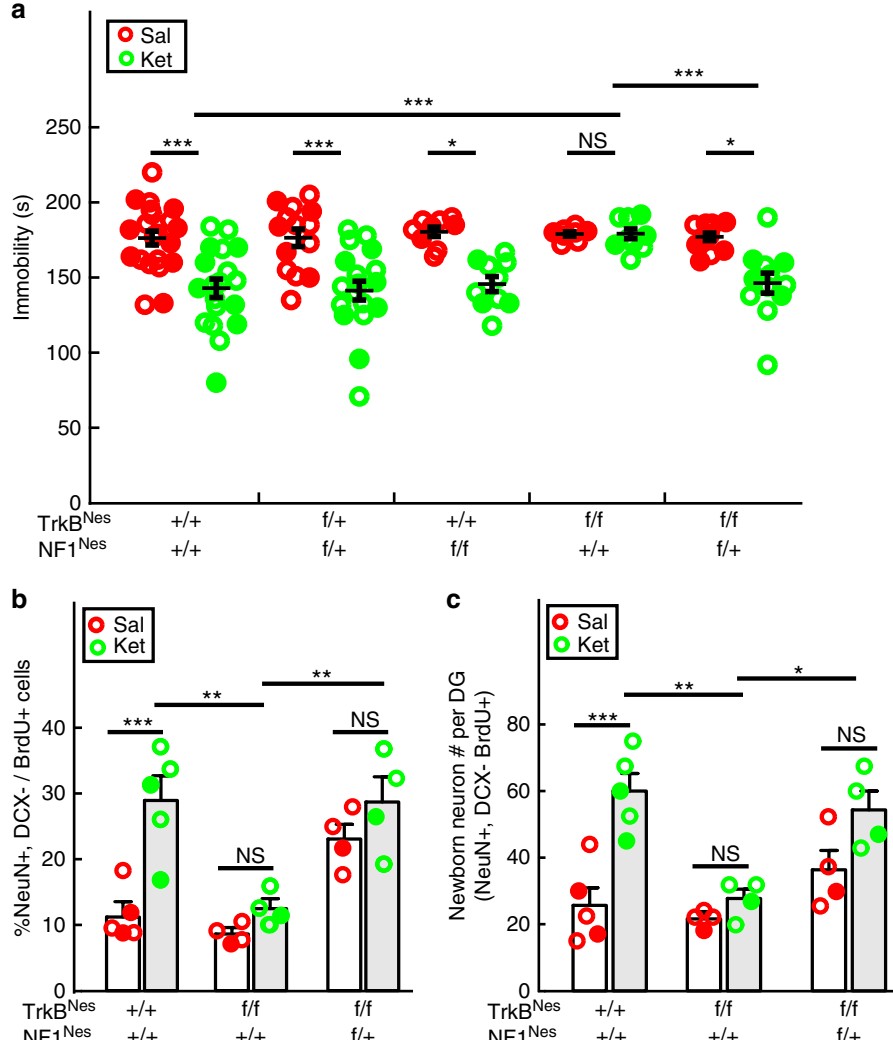

**Fig. 8** NF1 ablation rescues both neurogenic and behavioral deficits in mice with lacking TrkB in NPCs. **a** Ketamine significantly decreased immobility time in the FST in control, NF1Nes/TrkBNes double heterozygotes and NF1Nes mice, but not in TrkBNes mice. NF1Nes heterozygous (f/+); TrkBNes double-mutant mice reversed the impaired response to ketamine, as displayed by a significant reduction of immobility time in the FST (two-way ANOVA with unweighted mean analysis, Tukey post hoc, ***$p = 1.664E-5$ for ketamine-treated TrkBNes vs control mice; ***$p = 3.15E-4$ for ketamine-treated TrkBNes vs TrkBNes; Nf1Nes f/+; other comparisons were made between saline and ketamine within each genotype group). **b** Percentage of newborn neurons in ketamine-treated TrkBNes mice was significantly decreased compared to ketamine-treated control mice. In contrast, TrkBNes; NF1Nes f/+ mice displayed enhanced mature neuron generation compared to TrkBNes mice (two-way ANOVA, Tukey post hoc, ***$p = 0.0001$ for saline vs ketamine in control mice; **$p < 0.01$ for ketamine-treated TrkBNes vs control ($p = 0.0012$), or vs TrkBNes; NF1Nes f/+ mice ($p = 0.0081$); not significant for saline vs ketamine in TrkBNes mice ($p = 0.8555$) and TrkBNes; NF1Nes f/+ mice ($p = 0.6792$)). **c** Production of newborn neurons was rescued in ketamine-treated TrkBNes; NF1Nes f/+ mice compared to TrkBNes mice (two-way ANOVA, Tukey post hoc, ***$p = 0.0004$ for saline vs ketamine in control mice; **$p = 0.0014$ for ketamine-treated TrkBNes vs control, *$p = 0.0145$ for ketamine-treated TrkBNes vs TrkBNes; NF1Nes f+ mice; not significant for saline vs ketamine in TrkBNes mice ($p = 0.9749$) and saline vs ketamine in TrkBNes; NF1Nes f/+ mice ($p = 0.1674$)). Circles in solid color denote the data points from female mice

response to ketamine, but not the fast-acting behavioral effect that remains intact in TrkBNes mice with neurogenesis deficiency (Fig. 5c & Supplementary Fig. 7h). Whether adult neurogenesis sustains ketamine action in the stressed animal was not directly tested. However, given that ketamine and adult-born neurons behaviorally regulate the stress response[8,13,18,38], it is likely that adult neurogenesis is also required for ketamine-induced recovery of a depression-like state. Thus, ketamine-induced BDNF/TrkB signaling, not only modulates synaptic circuitry in the hippocampus, but also regulates adult hippocampal neurogenesis, both of which may work together to sustain ketamine's anti-depressive response.

The survival of DCX+ neural progenitors in the SGZ is regulated by neurotransmitters and neurotrophins[22,39]. Chronic treatment with conventional antidepressants promotes neural progenitor cell survival through enhanced BDNF signaling. Interestingly, our results showing that a low dose of ketamine mediates accumulation of newborn DCX+ progenitors not accounted for by progenitor proliferation and production suggests a survival effect on NPCs (Supplementary Fig. 1). However, this cellular effect that leads to the generation of new neurons and rewiring of neuronal circuits might be more likely to contribute to more latent behavioral responses rather than to the rapid action examined here.

The RAS-ERK pathway is critical for neurotrophin-mediated neuronal survival, differentiation, and activity[22,40]. In the brain, dysfunction of ERK signaling has been associated with impaired neural development and deficits in learning and memory[41]. Here

we add to the repertoire of ERK functions as a mediator of hippocampal progenitor cell differentiation. Although mTOR activation in prefrontal cortical neurons has been suggested to underlie the acute (1–2 h) action of ketamine[11], it is likely that different cell types or brain regions use selective signal transduction elements in the rapid response to ketamine. In addition, the transient ERK response ( < 24 h) coincides with the ketamine-induced temporal change in BDNF protein level in the DG, suggesting a BDNF dependent ERK signaling cascade. This view is supported by the studies of TrkB ablation that resulted in a block of ERK activation and anti-depressive response to ketamine. We also observed differential ERK activation between immature and mature DCX+ progenitors in response to ketamine, which is consistent with the long-term survival effect on NPCs and also echoes previous findings that more proliferative progenitors have higher TrkB expression compared with mature progenitors that have exited the cell cycle[20,33].

The *Nf1* gene encodes a tumor suppressor that through RAS-GAP activity, functions as a negative regulator of the RAS pathway[42]. Although NF1 can regulate both the ERK and AKT pathways, in hippocampal NPCs and their progeny, NF1[Nes] mice exclusively display ERK activation[21,43]. ERK activation via NF1 ablation in NPCs enabled hippocampal progenitors to accelerate differentiation into mature granule neurons, which not only prolonged the behavioral response to ketamine but also restored both the neurogenic and behavioral impairments observed in the TrkB[Nes] mice. The results support previous studies showing that ERK signaling is tightly linked to depression etiology and anti-depressant treatment[21,44]. However, enhanced neuronal differentiation in itself is not sufficient to achieve acute anti-depressive-like behavior in our model system. The NF1[Nes] mice display constitutively enhanced neurogenesis but did not show antidepressant-like behavior until treated with ketamine (Figs. 7e, f). Therefore, this neurogenic response likely contributes to the effect together with other mechanisms that underlie ketamine anti-depressive actions[9,10].

Chronic exposure to conventional antidepressants like SSRIs enhances neurogenesis by promoting stem cell entry into the cell cycle and enhancing NPC proliferation. One attractive explanation for the delay in anti-depressive action of conventional anti-depressants is the time required to gradually increase BDNF, which then stimulates neural stem/progenitor cell proliferation and eventual production of functional newborn neurons, ultimately leading to behavioral effects[12,20]. Ketamine instead accelerates the process through shortcuts involving rapid BDNF translation to promote the generation of functional newborn neurons within 24 h to sustain the AD response. We speculate that the acute or gradual stimulation of BDNF/TrkB signaling may have distinct effects on the regulation of neural stem/progenitor cell behavior (Supplementary Fig. 10). This idea is partially supported by previous in vitro studies that acute and gradual modulation of BDNF signaling produces opposing effects on hippocampal neuron activity owing to the differential kinetics of TrkB activation and downstream signaling including ERK pathway[45], suggesting that temporal aspects of BDNF/TrkB signaling may be crucial for regulating neural stem/progenitor cell biology. Understanding of the timing and scope of BDNF action upon induction by antidepressants will still require additional study. It will further be of interest to investigate the neurogenesis related mechanistic underpinnings for the ketamine metabolite, (2 R, 6 R)-hydroxynorketamine (HNK), recently reported to produce both rapid and sustained AD effect in an NMDAR inhibition-independent manner[46]. This effect may also be mediated by BDNF signaling and neural progenitor differentiation/maturation as described here for the sustained ketamine antidepressant effect.

## Methods

**Mice.** TrkB[flox] mice[34], NF1[flox] mouse line[42], Rosa26-DTA mice (The Jackson lab, stock# 006331)[28] and Nestin-creER[T2] transgenic mice[20,23] were generated as described previously. Rosa26-LacZ, YFP, and tdTOM reporter lines are from JAX (#003474, 025109 and 007914, respectively).All animals in these studies were maintained on 129/SvEv and C57BL/6 mixed background. TrkB[Nes] or NF1[Nes] mutant mice were generated by interbreeding TrkB flox/flox or NF1 flox/flox mice with TrkB[Nes] flox/flox or NF1[Nes] flox/flox mice, respectively. TrkB[Nes]; NF1[Nes] double-mutant mice were generated from the interbreeding between TrkB[Nes] flox/+; NF1[Nes] flox/+ double heterozygotes mice. Only DTA[Nes] heterozygote mice were used in the experiments and generated by breeding DTA[R26] with Nestin-creER[T2]. Control animals used in the experiments were either tamoxifen-induced mice that did not have the Nestin-creER[T2] transgene, or oil vehicle treated mice without DTA[R26], TrkB[flox] and Nf1[flox] alleles. Animals were grouping housed (2–4 mice/cage) on a 12-h light–dark cycled animal facility. All animal experiments utilized adult mice (between 10 and 16 weeks old) and analyzed with the experimenter blind to both genotype and treatment. All mouse protocols were approved by the Institutional Animal Care and Use Committee at the University of Texas Southwestern Medical Center at Dallas.

**Drug treatment.** Tamoxifen (Sigma) (500 mg/kg) in sunflower oil was administrated to adult mice by oral gavage. All other drugs were injected intraperitoneally (i.p.). Ketamine (7 mg/kg, Pfizer) was dissolved in saline and injected either 24 h or 1 week before the behavioral tests. The MEK inhibitor, SL327 (Sigma) was dissolved in DMSO (19 mg/ml) and injected (38 mg/kg) 1 h before saline/ketamine treatment[36]. The volume of each single injection was 60–80 μl. Two hours before the killing, a single i.p. injection of Kainate (35 mg/kg) was given to adult mice to induce grade 4–5 seizures and only mice displaying the rearing behavior would be included in the experiment[24]. For pulse-chase experiments, 4 doses of BrdU (5-Bromo-2'-deoxyuridine) were administered (100 mg/kg per injection, 2.5 h interval). To suppress adult neurogenesis, Temozolomide (TMZ, Sigma) was given to mice at 25 mg/kg (i.p., 2.5 mg/ml in saline), and the injections were performed on the first three days of a week for 4 weeks, followed by the forced swim test (see below) 4 weeks afterward[47] (Fig. 3a).

**Immunohistochemistry and quantitative analyses.** Mice were intracardially perfused with PBS followed by 2% paraformaldehyde (PFA) in PBS and the dissected brains postfixed in 4% PFA overnight at 4 degrees. Brains were then equilibrated in 30% sucrose and mounted in OCT (Fisher). Frozen sections were cut coronally at 18 μm. Immunofluorescence was performed as described previously[20,48]. Sections were incubated with the primary antibodies in blocking solution (10% donkey serum and 0.4% Triton X-100 in PBS) at the following dilutions: NeuN (1 : 200, Millipore, ABN78), Doublecortin (DCX) (1 : 200, Santa Cruz, sc8066), GFAP (1:500, BD bioscience 556329), Ki67 (1 : 200,Thermo, RM9106), Calbindin (1 : 200, Swant, #300), β-galactosidase (1 : 200, Abcam, ab9361) followed by the incubation with the corresponding donkey secondary antibodies conjugated with Cy2, Cy3 or Cy5 (1 : 300, Jackson Immunoresearch) and DAPI (Vector Labs) for counterstaining. For the staining of p-AKT (1 : 500, Cell signaling 9271), p-ERK (1 : 500, Cell signaling 4370), p-S6 (1 : 500, Cell signaling 2211), biotin-conjugated donkey anti-rabbit was used as secondary antibody and the fluorescent signal was detected using a tyramide signal amplification (TSA) kit (Perkin Elmer) following the manufacturer's instructions. For BrdU staining, sections were pretreated with 2 N HCl for 20 min to denature DNA for the exposure of BrdU antigen, followed by incubation with BrdU antibody (1 : 200, Abcam, ab6326) in blocking solution overnight. The representative images of immunofluorescence were taken from at least five animals.

Digital fluorescent images were obtained with a Nikon epifluorescence microscope and Zeiss LSM 710 confocal system. All representative confocal images are shown as a maximum intensity projection of a z-series stack acquired from 16 μm thick sections with a z-step of 2 μm. Every sixth coronal section from part of adult dentate gyrus (bregma: −2.46 to −1.34 mm) was used to assess neurogenesis in both hemispheres (10 sections per animal in each condition). Quantification of newborn neuron number/percentage (NenN+/DCX–/BrdU+ or NeuN+/DCX –/tdTOM) and DCX+ progenitors (DCX+/BrdU+ or DCX+/tdTOM) was performed on the orthogonal view of each confocal z-stack image for the precise determination of protein co-expression (total 50–60 BrdU+ or over 200 tdTOM+ cells were analyzed per animal), by using MetaMorph image analysis software (Molecular Devices)[20,48]. Many of DCX+ immature neurons have very low NeuN expression and were counted as DCX+ progenitors. Fluorescent images of DCX, Ki67, BrdU or GFAP positive Cells were captured in both hemispheres and counted manually and blindly for the genotypes or treatments. Cell counts from eight sections were multiplied by 6 to estimate the total number in every dentate gyrus. To quantify EGR1 expression in newborn neurons, 20–30 BrdU+ cell per animal were analyzed for the triple positive cell number and their percentage in newborn neurons. For the quantification of the DCX+ cell with dendrites in TMZ experiments, cells with dendrites longer than the diameters of DCX labeled soma were counted on the confocal images with maximum intensity projection of a z-series stack, and at least 8 images were quantified for each animal in both conditions. For the analysis of dendritic morphology, biocytin+ neurons or tdTOM + neurons with mostly intact dendritic branches were selected and 30 μm thick z-

series of images were collected for the dendritic tracing by NeuroJ. The number of secondary/tertiary dendrites and branch points was counted manually[21]. Fluorescent confocal images of DCX and p-ERK were captured from 16 μm thick sections and analyzed with the single confocal z-stack planes ($n = 4$ mice for each treatment). For experiments that include both male and female mice, we attempted to have similar gender proportions, which are indicated for comparison.

**Western blotting**. The dentate gyrus and CA1 region were carefully dissected from four hippocampal slices per mouse (~ 1mm thick). Tissue samples were rapidly frozen and lysed in RIPA buffer (Thermo Fisher Scientific). Protein samples from at least three animals were used independently for immunoblotting with antibodies at the following dilutions: BDNF (1 : 200, Santa Cruz, sc546), p-TrkB (1 : 1000, Abcam, ab81288), p-AKT (1 : 1000, Cell signaling 9271), p-ERK (1 : 1000, Cell signaling 9101), p-S6 (1 : 1000, Cell signaling 2211), ERK (1 : 1000, Cell signaling 9102), β-Actin (1 : 2000, Santa Cruz, sc69879). Western lightning plus-ECL (Perkin Elmer) was used for signal detection, and each blot might be stripped and reprobed with different antibodies. Quantification of band intensity normalized to the β-actin band was performed using MetaMorph image analysis software (Molecular Devices)[48].

**Acute dentate gyrus slice preparation**. Three weeks after the tamoxifen induction, adult mice (Nestin-creER$^{T2}$; tdTOM) were deeply anesthetized and transcardially perfused with chilled (4 °C) dentate gyrus dissection buffer containing (in mM): 110 Choline Cl, 2.5 KCl, 1.3 KH$_2$PO$_4$, 25 NaHCO$_3$, 10 D-glucose, 0.6 Na Pyruvate, 1.3 Na Ascorbate at pH 7.4, 300 mOsm, and aerated with 95% O2/5% CO2, including 0.5 CaCl2, 7 MgCl2, and 5 Kynurenate acid. Transverse hippocampal slices (250 μm thickness) were obtained on a Leica VT1200S slicer, and then recovered for 30 min at 35 °C and for another 15 min at room temperature in artificial cerebrospinal fluid (ACSF) containing (in mM): 125 NaCl, 2.5 KCl, 1.3 KH$_2$PO$_4$, 25 NaHCO$_3$, 1.3 Na Pyruvate, 10 D-glucose, 1.3 Na Ascorbate, 2 CaCl$_2$, 1.3 MgCl$_2$ aerated with 95% O2/5% CO$_2$ to pH 7.4. A single slice was then transferred to a submersion chamber and perfused at 3 ml/min with aerated ACSF at 30 °C ready to be recorded.

**Electrophysiology**. Lineage-traced cells in the dentate gyrus were identified under visual guidance using IR-DIC optics and tdTomato fluorescence, followed by the whole-cell patch clamp recordings. The recordings were performed using glass pipettes (3–5MΩ) filled with intracellular solution (0.2 mM EGTA, 130 mM K-gluconate, 6 mM KCl, 3 mM NaCl, 10 mM HEPES, 4 mM ATP-Mg, 0.4 mM GTP-Na, and 14 mM phosphocreatine-Tris at pH 7.2 and 285 mOsm) supplemented with 0.5% biocytin (B4261; Sigma) for the labeling. All recordings were obtained with a MultiClamp 700B amplifier. Currents were filtered at 2 kHz, acquired and digitized at 10 kHz using Clampex 10.3 (Molecular Devices). Action Potentials (APs) were recorded in current-clamp mode and elicited by a series of current injections ranging from −20 to 200 pA with 5, 10, or 20 pA increments and 800 ms duration. Sodium and potassium currents were recorded in voltage-clamp in response to a series of voltage steps ranging from −60 mV to +60 mV at 10-mV increments and 200 ms duration. Sodium current was measured as the inward current. Delayed rectifier potassium currents ($I_d$) were measured 50 ms prior to the end of the current step. Fast potassium currents ($I_A$) were measured at the peak of outward current immediately after sodium currents and then subtracted by $I_d$. Spontaneous synaptic currents were recorded in voltage-clamp mode. In all voltage-clamp recordings, cells were clamped at −60 mV except during the voltage-step protocol. In all current-clamp recordings, recordings were made at the resting membrane potential or without any current injection. Series and input resistance were measured in a voltage-clamp mode with a 400 ms, −10 mV step from a −60 mV holding potential (filtered by 10 kHz, sampled at 50 kHz). Cells were accepted only if the series resistance was less than 30 MΩ and stable throughout the experiment.

Data analysis was performed in Clampfit 10.3 software (Molecular Devices). The action potential (AP) trace immediately above threshold was used to determine the delay of 1st spike as the length of time from the start of current steps to the peak of AP; to measure AP threshold as the corresponding voltage when there was the sharpest change of trace slope; to determine AP amplitude, half width, maximum velocity of rise and decay by using the "Statistics" function from the "Analyze" menu as peak amplitude, half-width, maximum rise and decay slope. AP frequency was obtained by dividing the maximum number of spikes during the current steps protocol with the step time duration (800 ms). Similarly, sodium and potassium currents were measured using the "Statistics" function. The biggest current was used. Spontaneous postsynaptic currents and spontaneous endplate currents were analyzed by using the Mini Analysis Program v6.0.7 (Synaptosoft). Traces were imported into the program, and baseline noise was measured by "measuring RMS noise" option in "Noise Analysis" menu. Four times the noise was used as the threshold for detection. Each individual event was carefully examined to avoid false positives. Statistics can be shown from the "Events" menus.

**Behavior tests**. All behavioral experiments were performed during the light cycle. All animals used for the tests were littermates with all possible genotypes. The age of the mice in all experimental groups was 10–16 weeks and the age difference in each experimental group is within 2 weeks in order to eliminate any potential age-related influence on neurogenesis. Male (70%) and female (~30%) animals were tested. Since separate analyses show no statistically significant gender difference in each group, both data sets were incorporated. Researchers were blind to animal genotype and treatment information during testing.

The forced swim test (FST) was performed as described previously[20,21]. Since the test is sensitive to both conventional antidepressants and ketamine[8,49], it was used as the primary test for assessing the ketamine anti-depressive behavioral response in this study. Briefly, mice were placed in a 4 L glass beaker with 3 L water at 22–24 °C. Water was changed between subjects. All tests were recorded by a digital camcorder positioned on the side of the beaker. The videos were analyzed and scored by an experimenter blind to genotype/treatment and only the last 4 min of the total 6 min test was scored for immobility time. Immobility was defined as floating or remaining motionless except essential motions for keeping the head above the water. A decrease in immobility time indicates an antidepressant-like effect.

In the novelty-suppressed feeding test (NSFT), group-housed mice were deprived of food for 20 h prior to the test. Individual mice were placed in a 23 × 23 × 10 inch open field arena. Two or three pellets of food chow were placed in the center of the arena. The latency for each animal to approach and eat food was measured. Each test was terminated when the mouse first ate the food or when 10 min had elapsed without the mouse eating. Immediately following the NSFT, the amount of food consumed within 5 min in a normal housing cage was monitored as a control measure for appetite. Six days after NSFT, the same mice were subjected to the FST as an assessment of anti-depressive behavioral effect 1 week after the treatment.

In the learned helplessness test (LH), mice were individually placed on one side of a two-chamber shuttlebox (Med Associates) with the door closed for 1 h, receiving 120 shocks (0.35 mA for 2 s) with the unpredictable interval between shocks (18–44 s, average 30 s). This training was repeated for two days. On the test day, the door between the two chambers was raised at the onset of each shock and the shock ended either when the mouse stepped through the door or after 25 s. The number of escapes and the latency of each escape was recorded for 15 trials. Mice were not subjected to any other behavioral tests after LH.

**Statistical analyses**. All data are presented as mean ± SEM. Statistical analysis was done with GraphPad Prism or SPSS software. The D'Agostino & Pearson or Shapiro–Wilk normality test was first done on the data to determine whether it was a normal distribution. The sample size with continuous variables is calculated by SPSS. Statistical significance was defined as $p < 0.05$ using either a two-way ANOVA for comparisons among multiple groups (Supplementary Table 1) or an unpaired two-tailed $t$-test or the Mann–Whitney test for two groups of samples. Post hoc group multiple comparisons were performed with Turkey test. For two-way ANOVA with unequal sample size, the unweighted mean and type III Sum of Squares were calculated by SPSS. NS label in charts represents not significant in statistical analyses. One, two and three stars (*) in charts represent $p$-value < 0.05, < 0.01 and < 0.001, respectively.

**Data availability**. The data that support the findings of this study are available from the corresponding authors upon reasonable request.

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

## Acknowledgements

We thank Lauren Peca and Yanjiao Li for technical support, and all members of the Parada laboratory for valuable discussions and suggestions. We also thank Drs Lisa Monteggia, and Yun Li for critical reading of this manuscript and insightful comments. This work was supported by The Excellence in Education Fund of the Southwestern Foundation, by the Kent Waldrep Foundation for Nerve Regeneration, by the Southwestern Ball Chair and by the Diana and Richard C. Strauss Chair in Developmental Biology at UT Southwestern Medical Center.

## Author contributions

Z.M. and L.F.P. conceptualized the research and wrote the manuscript. Z.M. designed the experiments. Z.M., S.G.B., and Z.W performed molecular, cellular and behavioral studies. T.Z. and C-L.Z. performed electrophysiology experiments. J.E.J. provided reagents. Z.M. and T.Z. analyzed the data.

## Additional information

**Competing interests:** The authors declare no competing financial interests.

