## [Peer Review File · Nature Communications]

Reviewers' comments:

Reviewer #1 (Remarks to the Author):

The authors have done a good job addressing most of the concerns.

Two concerns remain:

1. To address the reviewer's concerns regarding specificity of driver line, please include coronal sections of ventral hippocampus (including CA1 and CA3) of Nestin CreERT2-LSTOPLYFP as previously requested.

2. The electrophysiological and morphological analysis suggest that the new neurons in the ketamine and control groups are equivalent (for eg: Input resistance and spiking properties). Isn't this surprising given that the authors assert that maturation of new neurons is enhanced/accelerated?

Reviewer #2 (Remarks to the Author):

The authors have thoroughly revised the manuscript and answered the points raised in the initial review.

The authors state that in the behavioral studies (line 636) both male and female mice were utilized and grouped together for analysis. Was this the case for all the other experiments too? It would be informative for the reader if the authors could report subject gender (maybe by color coding) in the figures.

Reviewer #1

1. *To address the reviewer's concerns regarding specificity of driver line, please include coronal sections of ventral hippocampus (including CA1 and CA3) of Nestin CreERT2-LSTOPLYFP as previously requested.*

As requested, in this revision, we provide additional Nestin-Cre expression data as coronal sections of ventral hippocampus including DG, CA1/CA2 and CA3 regions in Supplementary Fig. 4. Immunostaining of β -gal was used to indicate Cre recombinase expression in both ventral (sFig.4b, c) and dorsal (sFig.4d, e, f) hippocampal regions in Nestin^{creERT2}-LacZ mice. β -gal⁺ cells are restricted to the SGZ and some newborn neurons (β -gal⁺, NeuN⁺) are detected in the granular layer, but rarely seen in the CA1/CA2 and CA3 regions. β -gal immuno-signal (NeuN⁻) is also present in the molecular layer (ML) that reflects synaptic connections formed on newborn neuron dendrites. So, in this study, the Nestin-Cre transgene is able to specifically target neural stem/progenitor cells in the adult brain. The figure legend has been revised and highlighted.

2. *The electrophysiological and morphological analysis suggest that the new neurons in the ketamine and control groups are equivalent (for eg: Input resistance and spiking properties). Isn't this surprising given that the authors assert that maturation of new neurons is enhanced/accelerated?*

Yes, our analyses show that in addition to the increased number, newborn neurons following ketamine treatment are also functionally mature and appear normal both morphologically and electrophysiologically compared with normally-born mature neurons in controls. So, ketamine induced newborn mature neurons may not necessarily display more advanced/enhanced properties in electrophysiology and morphology. But the point is that ketamine accelerates the differentiation from DCX⁺ neural progenitor to NeuN⁺ neurons and thus increases the number of these functional newborn neurons within 24 hours, which appears to be essential to achieve the sustained antidepressant effect of ketamine in our cellular and behavioral analyses of mouse models where newborn neuron number is modulated.

Reviewer #2

The authors state that in the behavioral studies (line 636) both and male and female mice were utilized and grouped together for analysis. Was this the case for all the other experiments too? It would be informative for the reader if the authors could report subject gender (maybe by color coding) in the figures.

This is also the case for some other experiments in the study including some immunostaining experiments. The bottom line is that, if the tests involve both male and

female mice, we always keep the similar gender proportion (~30% female mice) in the groups analyzed for comparison and also determine if there is a statistically significant gender difference in each group. Here we have made it clearer in the “Methods” part.

We agree with the reviewer’s suggestion that giving the gender information could be more informative to readers. To this end, we have distinguished the data points from different genders by using empty circles (male) or solid circles (female) in the figures that involve both genders. We also give the gender composition for each N number in the legends of behavior test charts. Accordingly, we have added the information about this in the “Legends” and “Methods” parts. All the additions have been highlighted in the revised manuscript as requested.

REVIEWERS' COMMENTS:

Reviewer #1 (Remarks to the Author):

The image provides in Fig S4b show labeling in the SLM. Are these entorhinal axon terminals? Is there recombination in the entorhinal cortex? Please include the EC from the same animal.

We wish to thank the reviewers for their time and valuable comments regarding our manuscript. Here we submit a revised version of the manuscript for publication in Nature Communications. Specific point-by-point responses to the reviewers are indicated below.

Reviewer #1

1. *The image provides in Fig S4b show labeling in the SLM. Are these entorhinal axon terminals? Is there recombination in the entorhinal cortex? Please include the EC from the same animal.*

As the requested, we have included a representative image of the entorhinal cortex in the supplementary Fig.4c. The image is from the same animal where cre recombination in other regions including the DG, CA1/2, CA3, and mPFC has been depicted. Accordingly, we have edited the legends to make the explicit description of this. We have analyzed several sections from the ventral to the dorsal direction and do not detect any recombination in the entorhinal cortex region. The labeling in the SLM could be just due to the very few recombined CA1 neurons, auto-fluorescence and/or non-specific binding of the x-gal antibody. Therefore, we believe that the vast majority of Nestin-cre recombination in our paradigm occurs in the neurogenic niche.